# Early-in-Life Serum Aldosterone Levels Could Predict Surgery in Patients with Obstructive Congenital Anomalies of the Kidney and Urinary Tract: A Pilot Study

**DOI:** 10.3390/jcm12051921

**Published:** 2023-02-28

**Authors:** Pierluigi Marzuillo, Pier Luigi Palma, Anna Di Sessa, Agnese Roberti, Giovanni Torino, Maeva De Lucia, Emanuele Miraglia del Giudice, Stefano Guarino, Giovanni Di Iorio

**Affiliations:** 1Department of Woman, Child and of General and Specialized Surgery, Università degli Studi della Campania “Luigi Vanvitelli”, 80138 Naples, Italy; 2Pediatric Urology Unit, “Santobono-Pausilipon” Children’s Hospital, 80129 Naples, Italy

**Keywords:** congenital anomalies of the kidney and urinary tract, urinary tract obstruction, posterior urethral valves, ureteropelvic junction obstruction, primary megaureter, aldosterone, renin, transient pseudohypoaldosteronism

## Abstract

The aim of the study was to evaluate whether serum aldosterone levels or plasmatic renin activity (PRA) measured early in life (1–3 months) could predict a future surgical intervention for obstructive congenital anomalies of kidney and urinary tract (CAKUT). Twenty babies aged 1–3 months of life with suspected obstructive CAKUT were prospectively enrolled. The patients underwent a 2-year follow-up and were classified as patients needing or not needing surgery. In all of the enrolled patients, PRA and serum aldosterone levels were measured at 1–3 months of life and were evaluated as predictors of surgery by receiver-operating characteristic (ROC) curve analysis. Patients undergoing surgery during follow-up showed significantly higher levels of aldosterone at 1–3 months of life compared to those who did not require surgery (*p* = 0.006). The ROC curve analysis of the aldosterone for obstructive CAKUT needing surgery showed an area under the ROC curve of 0.88 (95%CI = 0.71–0.95; *p* = 0.001). The aldosterone cut-off of 100 ng/dL presented 100% sensitivity and 64.3% specificity and predicted surgery in 100% of cases. The PRA at 1–3 months of life was not a predictor of surgery. In conclusion, serum aldosterone levels at 1–3 months could predict the need for surgery during obstructive CAKUT follow-up.

## 1. Introduction

Congenital anomalies of the kidney and urinary tract (CAKUT) are the most common cause of chronic kidney disease in childhood [1,2].

Obstructive CAKUT might determine irreversible kidney damage (KD). The current diagnostic approach through renal imaging is quite invasive in terms of both biological and economic costs. Moreover, the need and timing of intervention are often only made when KD has developed. Therefore, a strict follow-up, in which KD due to an obstruction can develop or progress, is often needed to recommend surgery [3].

In this framework, the identification of potential biomarkers for surgery before irreversible KD occurs or progresses is crucial to accurately select babies requiring early intervention. Moreover, the identification of biochemical biomarkers could allow a less invasive follow-up for infants compared with the current radiological approach.

A urinary tract obstruction increases not only the local pressure, but also the obstructive stress in the tubule-interstitial compartment resulting in decreased renal plasma flow due to afferent arteriolar constriction. The drop in the glomerular filtration rate determines the activation of the renin–angiotensin system, resulting in increased plasma renin activity (PRA) and aldosterone levels [4,5]. Moreover, the development of a tubular resistance to renin and aldosterone—as consequence of the tubular obstructive stress—can also determine an increase in both PRA and aldosterone levels [6].

Recent evidence indicates that a progressive increase in PRA could help physicians in the selection of patients with asymptomatic prenatally diagnosed hydronephrosis needing surgery [7].

In line with this evidence, reports describing a transient pseudohypoaldosteronism (TPHA) in patients with urinary tract obstruction are available [8]. TPHA is characterised by exaggerated mineralocorticoid responses with elevated renin and aldosterone levels due to urinary tract obstruction-related tubular resistance, resulting in hyperkalaemia, hyponatremia, and metabolic acidosis [6,9,10]. The hypothesis behind this study is that patients with suspected obstructive CAKUT could present a “subclinical” TPHA and that the serum aldosterone levels could be helpful in the identification of subjects needing surgery.

The aim of this study was to evaluate whether the serum aldosterone levels or PRA measured early in life (1–3 months of age) could predict later surgical intervention for obstructive CAKUT.

## 2. Methods

Between October 2019 and January 2020, 30 babies aged 1–3 months with suspected obstructive CAKUT were prospectively enrolled. The enrolment and clinical management of these patients was performed at the University of Campania Luigi Vanvitelli, while the surgical management was performed at AORN Santobono-Pausilipon. The ethical committee of the University of Campania Luigi Vanvitelli approved the study (approval n° 371/2018). Informed consent was obtained from the parents of all patients.

The inclusion criteria were: (i) aged 1–3 months at our first evaluation; and (ii) suspected obstructive CAKUT defined by postnatal kidney ultrasound (KUS) showing unilateral or bilateral anterior–posterior diameter of the pelvis (APDP) ≥ 15 mm or megaureter ≥ 7 mm. The exclusion criteria were: (i) urinary tract infections (UTI) before enrolment; (ii) the presence of congenital anomalies involving other organs or systems; (iii) the presence of vesico-ureteral reflux detected by voiding cystourethrogram; (iv) denied consent; (v) not returning for the follow-up evaluations.

After the exclusion of 3 patients with UTIs, 4 patients denying their consent, and 3 patients lost to follow-up, 20 patients were included. None of the enrolled patients presented with vesico-ureteral reflux nor with UTI after the enrolment (all of the patients after enrolment started antibiotic prophylaxis; please see the “clinical protocol” paragraph for more details).

As most patients with suspected obstructive CAKUT reached an accurate diagnostic definition at 12 months of age [11], and due to the availability of a follow-up of at least 24 months, we were able to correctly classify (obstructive vs. non-obstructive CAKUT) all of the enrolled patients.

### 2.1. Clinical Protocol

All patients underwent neonatal KUS. At 1–3 months of life, the patients repeated another KUS, and were subjected to Tc99mMag3 scintigraphy (Mag3S) and cystography. After the initial work-up, KUS was also performed 3 and 6 months later. In case of stability or an increase in dilation at these ultrasound evaluations, the patients underwent a new Mag3S 6 months after the first. On the other hand, if the patients presented with an improvement in the dilation, they underwent another KUS after 5 months and then biannually until reaching 2 years of age (Figure 1). All of the enrolled subjects underwent antibiotic prophylaxis for the first 6 months of life in the case of hydronephrosis with APDP ≥ 15 mm, for the first 12 months of life in the case of megaureter ≥7 mm, and until urethrocystoscopy in the case of posterior urethral valve diagnosis at voiding cystourethrogram.

### 2.2. Selection of Patients for Surgical Correction

Patients eligible for surgery were selected on the basis of the current guidelines [3]: patients with (i) APDP > 30 mm or split renal function (SRF) < 40% at first Mag3S or SRF with delta > 10% at subsequent Mag3S; (ii) megaureter diameter > 10 to 15 mm with SRF < 40% at first Mag3S or a SRF with delta > 10% at subsequent Mag3S or with increasing hydroureteronephrosis; (iii) evidence of posterior urethral valves (PUV) at cystography. 

### 2.3. Biochemical Exams

Blood and urine samples (drawn through a sterile bag) were collected at the time of the first Mag3S execution between 1 and 3 months of life (when we usually evaluate kidney function by creatinine measurement). The PRA and serum aldosterone levels were measured at enrolment in all patients and after 6 months from surgery in surgically managed patients.

The PRA (normal values < 7.8 ng/mL/h [12]) and serum aldosterone levels (normal values <19.6 ng/dL [13]) were measured by a single laboratory using radioimmunological and chemiluminescent commercially available assays (DiaSorin S.p.A., Saluggia, Italy), respectively.

Serum sodium, potassium, chloride, calcium, phosphorus, and uric acid levels, as well as spot urinary sodium, potassium, chloride, calcium, phosphorus, and creatinine levels, were measured by ARCHITECT *c*4000 (Abbott, Illinois, United States). The estimated glomerular filtration rate (eGFR) was calculated as previously described [14].

Urinary beta-2-microglobulin levels were measured by chemiluminescent assay (DiaSorin S.p.A., Saluggia, Italy).

The fractional excretion of sodium (FENa) and of potassium (FEK) were calculated by simultaneous measurement of urine and serum values.

### 2.4. Definitions

A lower urinary tract obstruction (LUTO) was defined by the presence of posterior urethral valves, while an upper urinary tract obstruction (UUTO) was defined by the presence of ureteropelvic junction obstruction or obstructive megaureter.

### 2.5. Statistical Analysis

Considering the pilot nature of the study, the sample size was calculated using the formula proposed by Viechtbauer et al. [15]. With an expected need of surgery in at least 40% of the patients meeting the inclusion criteria of the present study and a confidence interval of 95%, the minimum sample size required was 6.

We compared the clinical and biochemical characteristics of the patients classified on the basis of the need for surgery or not, and on the basis of the evidence of spontaneous improvement of LUTO and UUTO.

Spontaneous improvement was defined by failing to meet the criteria for surgery and by the reduction in urinary tract dilations during the follow-up.

Differences among the continuous variables were analysed with an independent-sample t test for normally distributed variables and with a Mann–Whitney test in the case of non-normality. Qualitative variables were compared using a chi-squared test. PRA and serum aldosterone at 1–3 months of life were evaluated as potential predictors of surgery during the follow-up by receiver-operating characteristic (ROC) curve analysis. The Youden index was used to identify the best cut-offs [16].

## 3. Results

The mean age at enrolment was 0.18 years ± 0.17 standard deviation score (SDS) while the mean age at last follow-up was 2.2 years ± 0.17 SDS. At 1–3 months of life, all of the enrolled patients presented normal serum sodium, potassium, chloride, and creatinine levels and normal renal tubular function.

The general characteristics of the 20 included patients are shown in the Table 1.

The patients undergoing surgery presented higher serum potassium levels (within the normal range) (*p* = 0.01) and lower FEK levels (*p* < 0.001) at first observation compared with those not undergoing surgery. Moreover, the patients undergoing surgery also presented lower serum uric acid levels (*p* = 0.02) and higher beta-2-microglobulin levels (but always within the normal range) (*p* = 0.04) compared with those not undergoing surgery (Table 1).

In total, 6 out of 20 patients (*n* = 4 hydronephrosis, *n* = 2 megaureter) did not meet the criteria indicating the need for surgery, while 14 (70%) needed surgery (*n* = 7 posterior urethral valves, *n* = 4 unilateral megaureter, *n* = 3 unilateral ureteropelvic junction obstruction). The mean age at surgical intervention was 0.93 years ± 0.3 SDS. The patients who underwent surgery during follow-up showed significantly higher levels of aldosterone at 1–3 months of life compared with those who did not require surgery (*p* = 0.006). No differences in PRA levels were found (Table 1).

After the classification of patients on the basis of spontaneous improvement of the urinary tract dilations, of UUTO, and of LUTO, the serum levels of aldosterone, potassium, uric acid, and FEK still significantly differed among the groups (Table 2).

The ROC curve analysis of the aldosterone for obstructive CAKUT needing surgery showed an area under the ROC (AUROC) curve of 0.88 (95% confidence interval, CI, = 0.71–0.95; *p* = 0.001) (Figure 2A). Based on the Youden test, the best aldosterone cut-off was 100 ng/dL. This cut-off showed sensitivity = 100%, specificity = 64.3%, positive predictive value = 86.7%, and negative predictive value = 100%. The AUROC remained significant when separately analysing patients with LUTO (AUROC = 0.95; 95% CI = 0.84–1.0; *p* = 0.007) (Figure 2B) and UUTO (AUROC = 0.86; 95% CI = 0.64–1.0; *p* = 0.03) (Figure 2C). The best cut-off of aldosterone was 120.2 ng/dL for the LUTO and 97.7 ng/dL for the UUTO.

All patients with early-in-life serum aldosterone levels ≥ 100 ng/dL needed surgery during the follow-up. In all cases, the levels of aldosterone normalised within 6 months of surgery.

On the other hand, the PRA at 3 months of life did not show a significant AUROC for obstructive CAKUT needing surgery during the follow-up (AUROC = 0.58; 95% CI: 0.34–0.84; *p* = 0.54). Moreover, also when separately analysing patients with LUTO (AUROC = 0.75; 95% CI: 0.47–1.0; *p* = 0.13) and UUTO (AUROC = 0.57; 95% CI: 0.24–0.89; *p* = 0.67), the AUROCs were not significant.

## 4. Discussion

This is the first study designed to prospectively investigate the utility of PRA and serum aldosterone levels—measured early in life (1–3 months of life)—in predicting the need for surgery due to obstructive CAKUT during follow-up.

Most cases of CAKUT spontaneously improve and only a few cases need surgical intervention [17,18]. Unfortunately, patients needing surgery are often only identified after the occurrence of KD. In fact, according to the European Society of Paediatric Urology guidelines, an SRF < 40% or a delta >10% of the SRF at the follow-up Mag3S—both indicating already-established KD—represent criteria for surgery [3]. For this reason, it could be crucial to timely identify patients needing surgery.

Noteworthy is the fact that both PRA and aldosterone levels have been regarded as early biomarkers for obstructive CAKUT because the obstruction at the tubulo-interstitial level determines the activation of the renin–angiotensin system, resulting in increased PRA and aldosterone levels [19,20]. In addition, the obstruction also determines tubular damage with epithelial cell apoptosis [21]. Accordingly, we found higher beta-2-microglobuline levels among patients who underwent surgical correction compared with those who did not. This further underlines the occurrence of a direct tubular involvement in patients with obstructive CAKUT. Beta-2-microglobulin, in fact, is a low-molecular-weight protein that passes throughout glomeruli into the urine and is reabsorbed by renal proximal tubules [22]. Therefore, in the case of tubular damage, urinary beta-2-microglobulin levels are increased due to the lack of reabsorption [23].

Several reports linking TPHA with obstructive CAKUT or urinary tract infections are available [8,9,10,24]. The common characteristics of TPHA are hyperkalaemia, hyponatremia, and metabolic acidosis associated with high levels of aldosterone [6,9]. Interestingly, in the present study, high aldosterone levels were found in patients with obstructive CAKUT needing surgery, but symptomatic TPHA was not detected in any of the patients. The serum potassium levels were higher (but within the normal range) among patients needing surgery compared to those who did not, and a trend indicating lower sodium serum levels was only found in the first group. Accordingly, lower FEK levels were found in patients requiring surgical correction compared with those who did not. This may confirm the hypothesis that patients with obstructive CAKUT may present subclinical TPHA, and that only in more severe obstructions or in case of genetic predisposition, the subclinical TPHA could become overt.

Evaluating the potential role of aldosterone as a predictor of surgery in patients with suspected obstructive CAKUT, we found that serum aldosterone levels measured early in life (1–3 months) significantly predicted the need for later surgery both in patients with LUTO and UUTO. Obviously, in the case of LUTO (posterior urethral valves), the diagnosis can be easily made using voiding cystourethrography. However, evidence indicates that a preoperative suspicion of posterior urethral valves at voiding cystourethrography was present in only 46% of non-toilet-trained patients with posterior urethral valves [25]. In this case, the serum aldosterone levels could help in the selection of patients to submit for urethrocystoscopy in cases of doubtful voiding cystourethrography.

Bajpai et al., found that a progressive increase in PRA in children with asymptomatic prenatally diagnosed hydronephrosis reflects obstructive stress in the tubulo-interstitial compartment and could help in the selection of patients with obstruction [7]. The predictive value of PRA in terms of the need for surgery was not confirmed in the present study. Unlike Bajpai et al., who serially measured PRA in 30 patients aged 1–18 months [7], only patients aged 1–3 months old were enrolled and a single blood sample at this age was collected in our series. This could explain the different results between the study conducted by Bajpai et al., and the present study. It is likely that, differently from aldosterone, PRA requires a series of measurements in order to provide a predictive role in obstructive CAKUT. However, both serial PRA [7] and early-in-life serum aldosterone level measurements appear to be promising predictors of obstructive CAKUT, as shown in the paper by Bajpai et al., and in the present study, respectively. However, in terms of the number of blood sample collections and the number of biochemical dosages, aldosterone might have a greater clinical feasibility as a single measurement.

At follow-up, the aldosterone serum levels persistently normalised in all patients subjected to surgery. In contrast, Marra et al., showed minor disturbances in potassium balance in 18 infants with congenital hydronephrosis, with persistent hyperkalaemia and elevated levels of aldosterone and PRA as long as 3 years after surgery [24]. The authors only enrolled patients showing symptomatic TPHA before the surgical correction with the selection of the most severe cases [24]. This might be responsible for the persistent biochemical anomalies [24].

The main limitation of the present study is the sample size. Another limitation is the CAKUT heterogeneity of the included patients. Moreover, the concentration of both aldosterone and PRA may differ between laboratories, and is dependent on time sampling. However, this study does not suggest the utilisation of aldosterone in clinical practice at the present time, but rather represents a proof of concept and preliminary observation paving the way for future multicentre studies in this intriguing research area.

In conclusion, patients with obstructive CAKUT undergoing surgery during follow-up showed significantly higher levels of serum aldosterone at 1–3 months of life compared with those who did not require surgery, with significant area under the ROC curve for surgery during the follow-up. The aldosterone cut-off of 100 ng/dL presented 100% sensitivity and 64.3% specificity and predicted surgery in 100% of cases. The PRA at 1–3 months of life was not a predictor of surgery. However, this only represents a proof of concept, and future multicentre studies are needed to confirm this preliminary data and to implement serum aldosterone measurement in the clinical management of obstructive CAKUT.

## Figures and Tables

**Figure 1 jcm-12-01921-f001:**
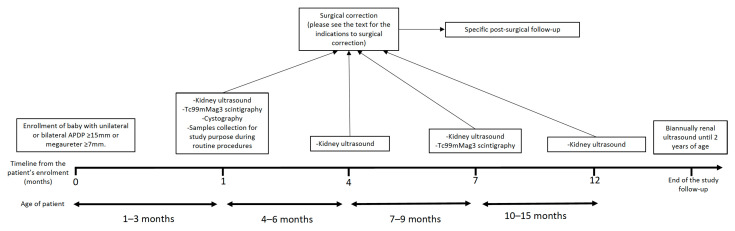
Protocol of the study.

**Figure 2 jcm-12-01921-f002:**
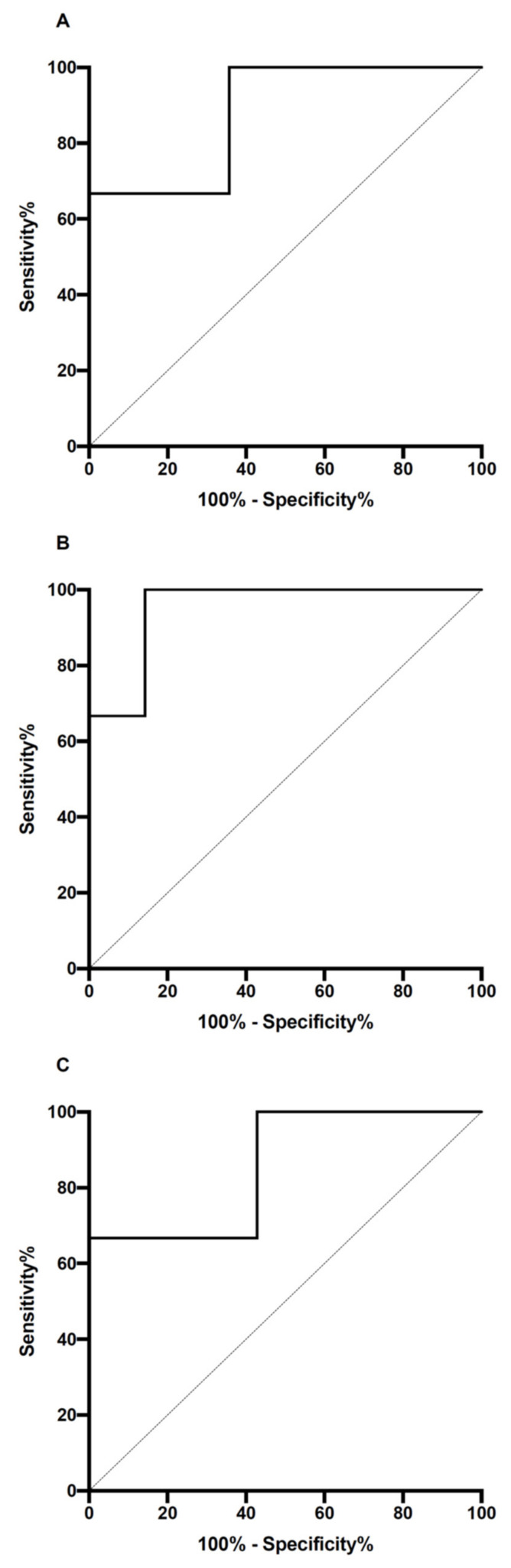
ROC curve analysis of aldosterone for obstructive CAKUT needing surgery. Panel (**A**), global population. Panel (**B**), patients with LUTO. Panel (**C**), patients with UUTO.

**Table 1 jcm-12-01921-t001:** Clinical and biochemical characteristics of patients presenting or not presenting spontaneous obstructive CAKUT resolution. For continuous variables, means ± SDS are shown. Abbreviations: FENa, fractional excretion of Na; eGFR, estimated glomerular filtration rate; FEK, fractional excretion of K; PRA, plasma renin activity; PUV, posterior urethral valves.

	Spontaneous Resolution No. = 6	Surgical Correction No. = 14	*p*
Age at first observation, yr	0.25 (0.32)	0.14 (0.05)	0.2
Male sex, No. (%)	4 (66.7)	11 (78.6)	0.57
Serum aldosterone, ng/dL	38.8 (40.1)	155.1 (86.8)	0.006
PRA, ng/mL/h	2.4 (1.1)	5.2 (9.1)	0.48
Sodium, mEq/L	138.2 (3.2)	137.0 (1.6)	0.3
Potassium, mEq/L	4.5 (0.9)	5.5 (0.6)	0.01
Chloride, mEq/L	104.6 (1.7)	105.4 (1.8)	0.4
Uric acid, mg/dL	2.77 (0.29)	2.33 (0.4)	0.02
Phosphorus, mg/dL	6.35 (0.66)	6.79 (0.72)	0.21
Creatinine, mg/dL	0.35 (0.05)	0.38 (0.09)	0.4
eGFR, mL/min/1.73 m^2^	97.11 (11.13)	93.04 (8.77)	0.39
FENa, %	0.69 (0.32)	0.6 (0.19)	0.4
FEK, %	26.66 (7.5)	14.16 (5.6)	<0.001
Beta-2-microglobulin, mg/L	0.05 (0.06)	0.20 (0.19)	0.04
Microalbuminuria mg/L	22 (7.4)	23.6 (14)	0.85
PUV, No. (%)	0 (0)	7 (50)	0.05
Megaureter, No. (%)	2 (33.3)	4 (28.6)	0.83
Hydronephrosis, No. (%)	4 (66.7)	3 (21.4)	0.12
Age at surgical intervention, yr	n.a	0.93 (0.3)	n.a
Age at last follow-up, yr	2.12 (0.12)	2.26 (0.17)	0.07

**Table 2 jcm-12-01921-t002:** Clinical and biochemical characteristics of patients classified on the basis of presence or no presence of spontaneous obstructive CAKUT improvement and on the basis of site of obstruction. For continuous variables, means ± SDS are shown. Abbreviations: FENa, fractional excretion of Na; FEK, fractional excretion of K; PRA, plasma renin activity.

	Spontaneous Improvement No. = 6	Upper Urinary Tract Obstruction No. = 7	Lower Urinary Tract Obstruction No. = 7	Global *p*
Age at first observation, yr	0.25 (0.32)	0.14 (0.03)	0.14 (0.06)	0.45
Male sex, No. (%)	4 (66.7)	4 (57.1)	7 (100)	0.83
Serum aldosterone, ng/dL	38.8 (40.1)	140 (110.40)	170.21 (60.01)	0.02
PRA, ng/mL/h	2.4 (1.1)	2.09 (1.16)	8.38 (12.42)	0.24
Sodium, mEq/L	138.2 (3.2)	136.71 (1.25)	137.29 (2.06)	0.52
Potassium, mEq/L	4.5 (0.9)	5.21 (0.85)	5.7 (0.51)	0.04
Chloride, mEq/L	104.6 (1.7)	105.29 (1.25)	105.43 (2.07)	0.70
Uric acid, mg/dL	2.77 (0.29)	2.16 (0.17)	2.5 (0.50)	0.02
Phosphorus, mg/dL	6.35 (0.66)	6.70 (0.67)	6.89 (0.80)	0.42
Creatinine, mg/dL	0.35 (0.05)	0.39 (0.13)	0.38 (0.03)	0.68
eGFR, mL/min/1.73 m^2^	97.11 (11.13)	95.22 (10.57)	90.85 (6.78)	0.49
FENa, %	0.69 (0.32)	0.53 (0.18)	0.66 (0.20)	0.42
FEK, %	26.66 (7.5)	12.08 (6.24)	16.24 (4.41)	<0.01
Beta-2-microglobulin, mg/L	0.05 (0.06)	0.18 (0.16)	0.21 (0.24)	0.22
Microalbuminuria mg/L	22 (7.4)	25.43 (19.34)	21.86 (6.39)	0.84

## Data Availability

The datasets generated and/or analysed during the current study are available from the corresponding author on request.

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
