# Peer review of "Early-in-Life Serum Aldosterone Levels Could Predict Surgery in Patients with Obstructive Congenital Anomalies of the Kidney and Urinary Tract: A Pilot Study"

_jcm, 2023, doi:10.3390/jcm12051921_

Round 1

Reviewer 1 Report

The authors report on a cohort of infants (n=20) with congenital anomalies of kidneys and urinary tract  (CAKUT) with high suspicion for obstructive uropathy (at either the lower or the upper urinary tract) seen over a relatively short period of time (in a single center?). In this cohort, different kidney function studies as well as serum renin and aldosterone levles were examined before a decision on the need for surgery for significant obstructive uropathy was done. Eventually, 14/20 infants needed surgery, 50% of them for PUV and only 3 for "hydronephrosis" (presumably UPJ stenosis). They report on significant higher aldosterone and potassium levels (but not PRA, contrary to previous publications quoted by them, such as  REF# 7) in the operated  vs the non operated group. However, the authors mix between LUTO (which is a clinical straight-forward diagnosis and is usually always operated, and Upper-UTO, which, especially if unilateral, still poses a difficult diagnostic challenge. Serum aldosterone and potassium levels are still significantly elevated in the U-UTO Vs control groups (Table 2 ), but the comparison is then performed  in a small number of patients (7 Vs 6 ).    Given the scarcity of studies examining RAAS components in non-UTI associated obstructive uropathy, this study is a sigificant contribution, in spite of the low number  of reported patients. 

Specific critique: 

1) Line 70, quoting REFS  # 4 and 5 : These 2 refs do not highlight this important statement by the authors: that urinary tract obstruction affects the tubulointerstitial compartment, then causing hemodynamic changes that lead to changes in the RAS.  

2) Line 72, REF #6: Again, this ref explains the source for aldosterone resistance in cases of UTI associated with CAKUT )obstructive and non obstructive).  

3) Line 135, Methods: Age related norms for serum aldosterone and PRA change very rapidly during infancy. What published norms were used bny the authors?

4)  Line 144: were FENa & FEK calculated by simultaneous measurement of urine and plasma values? This should be emphasized. 

5) Line 185: Please specify whther the obstructive megaureter and UPJ stenosis were uni- or bilateral . 

6) Discussion:  the authors should try to explain the discrepancy between their findings and the report in REF#7, where PRA levels were elevated before surgery in unilateral UPJ-O. This is contrary to the authors' curr ent report. 

Minor critique: 

a) Line 201: crrect to "Youden test"

Author Response

Reviewer 1

The authors report on a cohort of infants (n=20) with congenital anomalies of kidneys and urinary tract (CAKUT) with high suspicion for obstructive uropathy (at either the lower or the upper urinary tract) seen over a relatively short period of time (in a single center?). In this cohort, different kidney function studies as well as serum renin and aldosterone levels were examined before a decision on the need for surgery for significant obstructive uropathy was done. Eventually, 14/20 infants needed surgery, 50% of them for PUV and only 3 for "hydronephrosis" (presumably UPJ stenosis). They report on significant higher aldosterone and potassium levels (but not PRA, contrary to previous publications quoted by them, such as  REF# 7) in the operated  vs the non operated group. However, the authors mix between LUTO (which is a clinical straight-forward diagnosis and is usually always operated, and Upper-UTO, which, especially if unilateral, still poses a difficult diagnostic challenge. Serum aldosterone and potassium levels are still significantly elevated in the U-UTO Vs control groups (Table 2), but the comparison is then performed  in a small number of patients (7 Vs 6 ).    Given the scarcity of studies examining RAAS components in non-UTI associated obstructive uropathy, this study is a significant contribution, in spite of the low number  of reported patients. 

Answer: thank you. We specified the role of each center in the method section. The enrolment and clinical management of these patients has been performed at University of Campania Luigi Vanvitelli while the surgical management has been performed at AORN Santobono-Pausilipon. Please see lines 77-79 of the new version of the manuscript.

Specific critique: 

1) Line 70, quoting REFS  # 4 and 5 : These 2 refs do not highlight this important statement by the authors: that urinary tract obstruction affects the tubulointerstitial compartment, then causing hemodynamic changes that lead to changes in the RAS.  

Answer: we added the corrected original references. We previously cited two papers reassuming these findings but we agree with you that is more correct to add the original research firstly demonstrating this affirmation. Please see references 4 and 5 and line 58 of the new version of the manuscript.

2) Line 72, REF #6: Again, this ref explains the source for aldosterone resistance in cases of UTI associated with CAKUT )obstructive and non obstructive).  

Answer: we added the corrected original references We previously cited a paper reassuming this finding but we agree with you that is more correct to add the original research firstly demonstrating this affirmation. Please see reference 6 and line 60 of the new version of the manuscript.

3) Line 135, Methods: Age related norms for serum aldosterone and PRA change very rapidly during infancy. What published norms were used by the authors?

Answer: we added this information in the new version of the manuscript. Please see lines 126-127 of the new version of the manuscript.

4)  Line 144: were FENa & FEK calculated by simultaneous measurement of urine and plasma values? This should be emphasized. 

Answer: yes, the measurement was simultaneous. We added this information in the new version of the manuscript. Please see lines 136 and 137 of the new version of the manuscript.

5) Line 185: Please specify whether the obstructive megaureter and UPJ stenosis were uni- or bilateral. 

Answer: we specified that they were unilateral. Please see line 178 of the new version of the manuscript.

6) Discussion:  the authors should try to explain the discrepancy between their findings and the report in REF#7, where PRA levels were elevated before surgery in unilateral UPJ-O. This is contrary to the authors' curr ent report. 

 Answer: we added an explanation. Please see lines 255-261 of the new version of the manuscript.

Minor critique: 

  1. a) Line 201: correct to "Youden test"

Answer: we corrected the typo. Please see line 194 of the new version of the manuscript.

Author Response

Reviewer 2

It is a statistically correct study. The idea is not new, and it's value is rather experimental, since most of the obstructive CAKUT are diagnosed prenatally, and there are follow-up protocols and criteria for a surgical procedure.

Answer: thank you.

abstract. Much over 200 words. I suggest to keep the elements that objectively represent the article.

Answer: following your suggestion we reduced the length of the abstract to 200 words.

introduction

lines 57-59: In my personal opinion, the need and timing of intervention are quite good established based on imaging (US, VCUG, PIV, scintigrams). You may say that it is less invasive for an infant to be under biochemical surveillance.

Answer: we modified the text accordingly. Please see lines 46-53 of the new version of the manuscript.

Methods

How did you exclude the vesicoureteral reflux ?

Answer: We excluded vesico-ureteral reflux by voiding cystourethrogram. We added this information in the new version of the manuscript (please see line 87).

Discussion

line 282: at the present time but it represents only a

Answer: We modified the text accordingly. Please see line 278 of the new version of the manuscript.